# Acid Sphingomyelinase Deficiency Ameliorates Farber Disease

**DOI:** 10.3390/ijms20246253

**Published:** 2019-12-11

**Authors:** Nadine Beckmann, Katrin Anne Becker, Stephanie Kadow, Fabian Schumacher, Melanie Kramer, Claudine Kühn, Walter J. Schulz-Schaeffer, Michael J. Edwards, Burkhard Kleuser, Erich Gulbins, Alexander Carpinteiro

**Affiliations:** 1Department of Molecular Biology, University of Duisburg-Essen, Hufelandstraße 55, 45147 Essen, Germany; nadine.beckmann@uk-essen.de (N.B.); Katrin.Becker@uk-essen.de (K.A.B.); stephanie.kadow@uk-essen.de (S.K.); fabian.schumacher@uni-potsdam.de (F.S.); melanie.kramer@uk-essen.de (M.K.); claudine.kuehn@uk-essen.de (C.K.); erich.gulbins@uk-essen.de (E.G.); 2Department of Toxicology, Institute of Nutritional Science, University of Potsdam, Arthur-Scheunert-Allee 114-116, 14558 Nuthetal, Germany; kleuser@uni-potsdam.de; 3Insitute of Neuropathology, University of the Saarland, Kirrberger Str. 100, 66421 Homburg, Germany; walter.schulz-schaeffer@uks.eu; 4Department of Surgery, University of Cincinnati, 231 Albert Sabin Way, ML 0558, Cincinnati, OH 45229, USA; edwardm6@ucmail.uc.edu; 5Department of Hematology, University Hospital Essen, Hufelandstraße 55, 45147 Essen, Germany

**Keywords:** Farber disease, lysosomal storage disorders, acid ceramidase, acid sphingomyelinase, amitriptyline

## Abstract

Farber disease is a rare lysosomal storage disorder resulting from acid ceramidase deficiency and subsequent ceramide accumulation. No treatments for Farber disease are clinically available, and affected patients have a severely shortened lifespan. We have recently reported a novel acid ceramidase deficiency model that mirrors the human disease closely. Acid sphingomyelinase is the enzyme that generates ceramide upstream of acid ceramidase in the lysosomes. Using our acid ceramidase deficiency model, we tested if acid sphingomyelinase could be a potential novel therapeutic target for the treatment of Farber disease. A number of functional acid sphingomyelinase inhibitors are clinically available and have been used for decades to treat major depression. Using these as a therapeutic for Farber disease, thus, has the potential to improve central nervous symptoms of the disease as well, something all other treatment options for Farber disease can’t achieve so far. As a proof-of-concept study, we first cross-bred acid ceramidase deficient mice with acid sphingomyelinase deficient mice in order to prevent ceramide accumulation. Double-deficient mice had reduced ceramide accumulation, fewer disease manifestations, and prolonged survival. We next targeted acid sphingomyelinase pharmacologically, to test if these findings would translate to a setting with clinical applicability. Surprisingly, the treatment of acid ceramidase deficient mice with the acid sphingomyelinase inhibitor amitriptyline was toxic to acid ceramidase deficient mice and killed them within a few days of treatment. In conclusion, our study provides the first proof-of-concept that acid sphingomyelinase could be a potential new therapeutic target for Farber disease to reduce disease manifestations and prolong survival. However, we also identified previously unknown toxicity of the functional acid sphingomyelinase inhibitor amitriptyline in the context of Farber disease, strongly cautioning against the use of this substance class for Farber disease patients.

## 1. Introduction

Farber disease (FD) is a rare lysosomal storage disorder resulting from acid ceramidase (human AC, murine Ac) deficiency [1]. AC is a lipid hydrolase and deacetylates ceramide to sphingosine and free fatty acid. Ceramide is an important lipid mediator and has been implicated in a number of different cellular contexts, including apoptosis and inflammation (reviewed in [2,3]). AC deficiency in FD results in an accumulation of lysosomal ceramide [4], which is considered the cause of the disease.

FD clinically presents with a triad of deformed joints, subcutaneous nodules, and progressive hoarseness [5]. It can resemble juvenile idiopathic arthritis in infancy, which is why a number of FD patients are misdiagnosed at first [6,7,8]. FD is diagnosed by demonstration of reduced AC activity and abnormally high ceramide levels in cultured cells, biopsy samples, urine—or prenatally in amniocytes or chorionic villus samples [9,10,11,12,13,14,15]. Histologically, granulomas with lipid-laden macrophages are a characteristic feature of FD [16]. Seven subtypes of FD have been classified based on organ involvement and symptom-severity [5]. Subtypes I and IV include severely affected patients who rarely survive past two years of age and exhibit lung involvement (type I and IV), neurological deficits (type I) and hepatosplenomegaly (type IV). Subtypes II–III, V–VII include patients with a less severe phenotype (type II–III, V–VII), who may survive into adulthood. The variability of FD cannot be explained so far, as residual enzyme activity in an in vitro assay does not correlate with disease severity [5]. Overall, the pathophysiology is still poorly understood, in part owing to the very low incidence rate.

To shed light into the pathophysiological basis of FD, we and others have developed murine models of the disease. The earliest attempt to generate a knockout line targeted the catalytic domain of Ac and abrogated Ac activity completely. This resulted in early embryonic lethality [17]. Ac was subsequently shown to be upregulated during embryonic genome activation in order to overcome the ceramide-mediated default apoptosis pathway initiated in unfertilized oocytes [18]. The complete lack of Ac activity in the first knockout model prevented this, thus arresting further embryonic development. A second model introduced a point mutation into the *Asah1* gene, at a site that is putatively involved in binding of the co-factor saposin D [19]. A third model targeted the signal peptide of the enzyme, leading to a truncated protein that is not targeted to the lysosomes [20]. The latter two models retained sufficient residual Ac activity to overcome the apoptotic threshold during early embryogenesis and yielded viable offsprings with a variety of pathological manifestation of FD. These include severely shortened lifespan, tissue infiltration with lipid-laden macrophages, joint pathologies, perturbed hematopoiesis, changes in plasma cytokine levels, central nervous system abnormalities, pulmonary inflammation, visual impairments, hepatic fibrosis, muscular dystrophy, and reduced renal function [19,20,21,22,23,24,25].

Currently, the only option to ameliorate these symptoms in FD patients is supportive measures. No cure for FD exists to date. Allogenic haematopoetic stem cell transplantation has shown favorable results on joint manifestations [8,26]. Unfortunately, this option is unsuitable for patients with pulmonary disease and neurological involvement, which are hallmarks of severe FD [27]. The same shortcoming applies to AC enzyme replacement therapy, which is currently being developed and has obtained some promising results in cell culture- and mouse studies [28]. Thus, there are essentially no holistic treatment options for severely affected FD patients.

Since lysosomal ceramide accumulation is considered to be the cause of the disease, preventing ceramide generation in the lysosomes could be a new treatment strategy for FD. Lysosomal ceramide is generated by acid sphingomyelinase (human ASM, murine Asm) through hydrolysis of the abundant membrane lipid sphingomyelin. Indications that inhibition of ASM can successfully decrease and normalize ceramide accumulation come from studies on cystic fibrosis. In cystic fibrosis, pulmonary ceramides accumulate, contributing to inflammation, cell death, and infection susceptibility [29]. All of these pathologies were corrected by genetic or pharmacological inhibition of ASM [29,30]. Treatment of cystic fibrosis patients with the functional ASM inhibitor amitriptyline showed promising results in a phase II clinical study [31]. In light of these findings, we assessed the potential of ASM inhibition as a treatment for FD—as a proof-of-concept study, we cross-bred Ac-deficient mice to Asm-deficient mice and monitored ceramide accumulation, survival, and disease manifestations in FD mice upon co-ablation of Asm.

## 2. Results

### 2.1. Co-Deficiency of Asm Blunts Ceramide Accumulation in Ac-Deficient Mice

A key feature of FD is the accumulation of ceramide, which is also thought to be the cause of the disease. The murine models have consistently reported increased ceramide levels in Ac-deficient mice [19,20]. To test whether a deficiency of Asm in FD mice corrects the accumulation of ceramide, we crossed Ac-deficient mice with Asm-deficient mice to obtain double knockout mice. We then quantified splenic ceramide levels in the different mouse lines by liquid chromatography/mass spectrometry. We chose to analyze spleens since the spleen was among the organs with the highest total ceramide levels in Ac-deficient mice in our previous study [20].

Similar to our previous findings, total ceramide was approximately 130-fold elevated upon Ac deficiency compared to wildtype (Wt) mice of a similar age (Figure 1a). This ceramide accumulation spread over all tested ceramide species, independent of chain length (Appendix A). Heterozygous or homozygous knockout of Asm in Ac-deficient mice drastically decreased ceramide levels, particularly total ceramides as well as C16- and C24:1 ceramide (Figure 1a, Appendix A). Only C20 and C22 ceramide levels were not ameliorated by homozygous Asm co-deficiency. As the contribution of the individual ceramide species to FD manifestations is currently unknown, the relevance of C20 and C22 ceramide levels remaining elevated is currently unknown. Homozygous deletion of Asm also resulted in an increase of all tested sphingomyelin species in Ac-deficient mice to a similar extent as the increase of sphingomyelin observed in regular Asm-deficient mice (Figure 1b).

### 2.2. Co-deficiency of Asm Improves Weight Gain and Survival of Ac-deficient Mice

We tested the therapeutic relevance of the partially reduced ceramide levels upon Ac and Asm co-deficiency by monitoring body weight and survival. We previously reported that Ac-deficient mice weigh significantly less than their littermates already at the age of weaning, barely gain weight afterward, and start to lose weight around week 5, prior to dying at around week 6 [20]. Heterozygous knockout of Asm in Ac-deficient mice resulted in only slightly improved body weights and survival times compared to solely Ac-deficient mice, whereas homozygous deletion of Asm normalized weight gain in juvenile mice completely and delayed the onset of weight loss to an age of approximately 13 weeks (Figure 2a). The average survival time of the double deficient mice was 151 days, and thus, approximately three times longer than that of regular Ac-deficient mice (Figure 2b). We did not include regular Asm-deficient mice in these analyses since these mice develop Niemann-Pick disease and the competent authority (State Agency for Nature, Environment and Consumer Protection (LANUV) NRW in Düsseldorf, Germany), thus only allowed us to monitor these mice for up to 10 weeks. Asm-deficient mice showed normal weight gain, and no Asm-deficient mouse died during this 10-week observation time.

### 2.3. Asm Ablation in Ac-deficient Mice Ameliorates Histopathological Signs of FD

Histologically, the hallmark feature of FD is the presence of granulomas with lipid-laden macrophages. Heterozygous deletion of Asm ameliorated histiocytic infiltration in lymphoid organs of Ac-deficient mice, and homozygous deletion of Asm normalized the normal tissue architecture almost completely (Figure 3a–c). Asm co-deficiency, however, did not ameliorate pulmonary and hepatic inflammation (Figure 3a,d), but did improve liver fibrosis (Figure 3a,e) in Ac-deficient mice.

### 2.4. Asm Ablation in Ac-deficient Mice Improves the Cytokine Profile Characteristic for FD

The tissue infiltration with lipid-laden macrophages is thought to be driven by monocyte chemoattractant protein-1 (MCP-1) [32]. A characteristic serum cytokine profile has recently been described for FD patients, including increases of MIP-1alpha, IL-6, IL-12, VEGF, and IP-10 in addition to MCP-1 [22]. We previously verified increases of MCP-1, MIP-1alpha, VEGF, and IP-10 in our Ac-deficient mouse model. IL-6 remained unchanged and IL-12 levels were below the detection limit in our mouse model [20]. Hematopoietic stem cell transplantation (HSCT) as supportive therapy for FD has previously been reported to normalize MCP-1, IP-10, and IL-6 levels [22]. To test if Asm ablation would also lead to an improvement, we quantified MCP-1 levels in serum and spleens of our mice, as well as MIP-1alpha, VEGF, and IP-10 serum levels. As expected, based on the reduction of histiocytic infiltration, Asm co-deficiency significantly reduced MCP-1 levels in spleen and serum of Ac-deficient mice (Figure 4a,b). Additionally, Asm co-deficiency also significantly reduced serum MIP1-alpha (Figure 4c), and serum VEGF levels (Figure 4d). The increase in serum IP-10 upon Ac deficiency; however, was not improved by Asm co-ablation (Figure 4e).

### 2.5. Asm Ablation in Ac-deficient Mice Improves Clinical Parameters to Assess Overall Health

To further assess the overall health of Ac-deficient mice upon Asm co-ablation, we tested for more manifestations of FD that we previously observed in our model [20].

We reported that Ac-deficient mice have a number of hematological abnormalities, including reduced numbers of circulating leukocytes. Homozygous deletion of Asm in Ac-deficient mice normalized leukopenia (Figure 5a).

Blood urea nitrogen (BUN) levels are an indicator of kidney function. BUN levels are elevated in Ac-deficient mice ^20^. Here we show that homozygous co-depletion of Asm normalized this FD symptom as well (Figure 5b).

In addition, liver failure is a common complication of FD, which occurs in up to 25 % of patients [33]. Ac-deficient mice show strong elevations in liver transaminases, but normal serum protein, bilirubin, and albumin levels [20]. We, thus, determined glutamic-oxalacetic transaminase (GOT) and glutamate-pyruvate transaminase (GPT) in Ac-deficient mice upon Asm co-deficiency. The increase in liver transaminases in Ac-deficient mice was ameliorated substantially by Asm co-deficiency (Figure 5c,d).

We previously reported indications of muscular dystrophy in Ac-deficient mice. Additionally, mutations in the ASAH1 gene have also been described in a subset of spinal muscular atrophy patients [34]. We analyzed markers of (muscle) tissue damage, creatinine phosphokinase, and lactate dehydrogenase, which are both frequently elevated in spinal muscular atrophy patients and were elevated in our Ac-deficient mouse model as well [20]. Both LDH and CPK levels were significantly reduced already upon heterozygous deletion of Asm in Ac-deficient mice (Figure 5e).

### 2.6. Pharmacological Targeting of Asm with Amitriptyline Kills Ac-Deficient Mice

In light of the prolonged survival of Ac-deficient mice and the amelioration of many FD manifestations upon co-ablation of Asm, we wanted to test if similar results could be achieved by pharmacological inhibition of Asm. Amitriptyline is a functional inhibitor of Asm, which has been in clinical use since the 1960’s. We treated Ac-deficient mice with 180 mg/L amitriptyline in normal saline via the drinking water, to which the mice had *ad libitum* access. The fresh solution was prepared every three days. This regimen has previously been reported to inhibit Asm in vivo effectively [35,36]. We started treatment at four weeks of age, shortly before the onset of weight loss in Ac-deficient mice. Treatment with amitriptyline severely shortened the survival time of Ac-deficient mice compared to controls receiving just normal saline (Figure 6a). The death occurred within two weeks of treatment initiation, and the median survival time in the amitriptyline treated group was only 37 days. No adverse effects were seen on age-matched Wt mice upon treatment with either amitriptyline or normal saline (Figure 6a). We quantified Asm activity in amitriptyline treated Ac-deficient mice, but surprisingly only found a significant reduction in the liver (Figure 6b–d). Ceramide and sphingomyelin levels in the liver remained unaltered however, irrespective of the tested chain length (Figure 6e,f, Appendix A).

### 2.7. Amitriptyline Treatment is Toxic for Ac-deficient Mice

Given the shortened survival times of Ac-deficient mice treated with amitriptyline, we tested clinical parameters to gain further insights into the potential source of this toxicity. Amitriptyline treatment significantly elevated liver transaminases and lactate dehydrogenase, as well as blood urea nitrogen levels (Figure 7) in Ac-deficient mice. It also furthered leukopenia in Ac-deficient mice and led to a significant reduction in platelet counts (Figure 7). No such effects were seen in Wt controls treated with amitriptyline.

## 3. Discussion

FD is a severe genetic disorder with very limited treatment options. The development of genetic mouse models in recent years has helped gain further insights into the disease pathology and presented us with the opportunity to identify and test the efficacy of new treatment options.

We recently published the phenotype of one of the FD mouse models, targeted deletion of Exon 1 of the *Asah1* gene, which removes the signal peptide sequence of Ac. As a result, lysosomal targeting of the mutant Ac protein is disrupted, leading to reduced Ac activities and lysosomal ceramide accumulation [20]. FD manifestations in our mice closely mimic severe forms of FD in human patients, and also correspond to the phenotype described for the other viable FD mouse model that has been previously described [19]. FD manifestations in our model include failure to thrive, shortened survival, histiocytic infiltration in many tissues, liver disease, as well as indications of muscular dystrophy and renal disease [20].

Given the lack of treatment options for FD patients, we wanted to use our model to investigate a novel treatment strategy for FD. We chose to target lysosomal ceramide accumulation since this is thought to be the cause of FD. Additionally, other diseases are also associated with ceramide accumulation. For instance, the lung inflammation present in Ac-deficient mice matches findings in cystic fibrosis, in which pulmonary ceramide accumulates [29]. Hepatic ceramide accumulation has previously been linked to non-invasive hepatocellular carcinoma in a ceramide synthase 2 knockout model [37]. Therefore, a treatment strategy aimed at reducing ceramide accumulation also has numerous implications outside of the context of FD and potentially widespread clinical uses.

As a new treatment strategy for FD, we chose to target Asm, the lysosomal enzyme that generates ceramide from sphingomyelin upstream of AC. As a proof-of-concept study, we first used a genetic model: By crossbreeding Ac-deficient mice to Asm-deficient mice, we could show the suitability of Asm depletion as a new treatment strategy for FD and ceramide accumulation in general. Homozygous co-depletion of Asm significantly improved weight gain and survival of Ac-deficient mice and reduced histiocytic infiltrations in the lymphoid organs. Leukopenia, hepatic manifestations (liver fibrosis, GOT, and GPT elevations) and signs of renal disease and muscular dystrophy were also markedly improved. Heterozygous deletion of Asm was less effective and could only improve the median survival time by a few days. However, pharmacological inhibition of Asm will most likely achieve better reductions of Asm activity then heterozygous knockout and should, therefore, be clinically effective.

Despite prolonging the survival of Ac-deficient mice, the co-depletion of Asm did not cure FD completely in these mice and did not save them from dying. This is likely due to the fact that Asm co-depletion only reduced ceramide accumulation in Ac-deficient mice, but did not normalize ceramide levels completely. Additionally, Asm deficiency itself is known to cause Niemann-Pick disease due to sphingomyelin accumulation. Ac and Asm co-deficient mice showed similar sphingomyelin levels as regular Asm deficient mice in spleens. An increase in sphingomyelin has not been investigated in detail in FD patients so far, but one study has suggested an increase in 24:1 sphingomyelin in plasma [38], and we previously noted increases in total sphingomyelin and C16-sphingomyelin levels in lung, liver, spleen, kidneys, muscle, and brain tissue in our Ac-deficient mice [20]. Increased sphingomyelin levels, either from Ac or Asm deficiency, may contribute to FD manifestations like foam cell infiltration, pulmonary and hepatic involvement, and central nervous symptoms (reviewed in [3]). This is a potential downfall of targeting Asm in FD. However, known Asm inhibitors retain some residual activity of Asm, which, based on the clinical experience with these drugs so far, is sufficient to maintain sphingomyelin homeostasis but will also be more effective than a heterozygous genetic depletion of Asm. Based on our data in the genetic model, they would, therefore, likely have ameliorative effects on FD manifestations.

To test this, we targeted Asm pharmacologically in a second study. Several functional inhibitors of ASM are already in clinical use, e.g., the tricyclic antidepressants amitriptyline, desipramine, imipramine, and fluoxetine [39,40,41]. Their favorable properties include good absorption, distribution, metabolism and excretion, activity across different cell types, no habituation, reversible inhibition, and no rebound effects [40]. As they can also cross the blood-brain barrier, they would be the first treatment option for FD that could also target central nervous symptoms. For our study, we used amitriptyline and treated the mice via their drinking water starting at four weeks of age, about a week before they normally start to decline due to FD. While amitriptyline treatment had no adverse effects on Wt mice, Ac-deficient mice died within two weeks of treatment. This was in strong contrast to the favorable results we noted upon the genetic deficiency of Asm. Quantification of Asm activity in several organs of the treated mice surprisingly revealed that a significant reduction of Asm activity was only achieved in the liver. This was unexpected and not sufficient to impact ceramide and sphingomyelin levels. At this point, it is unclear why Asm inhibition in Ac-deficient mice was less effective than what has been reported in healthy controls. We speculate that Ac-deficient mice consumed less amitriptyline via their drinking water than healthy controls, perhaps due to their diseased state. It is also important to note that these data have a survivor bias since up to 40 % of Ac-deficient mice treated with amitriptyline die within the first seven days of treatment.

It is unclear why amitriptyline was toxic to Ac-deficient mice at this time. It is possible that the mechanism of Asm inhibition by amitriptyline is responsible: Functional Asm inhibitors like amitriptyline exert their Asm inhibitory effect by interfering with Asm membrane attachment and thus targeting Asm for proteolytic degradation [42]. Since Ac is attached to the lysosomal membrane in a similar fashion as Asm and even exists in a complex with Asm [43], functional Asm inhibitors may induce degradation of Ac as well. This has already been reported to be the case for desipramine in an in vitro study [44]. Further reduction of the residual activity of Ac in FD could explain the observed detrimental effect. In the Ac-deficient mice that were available for testing, amitriptyline treatment did not significantly change ceramide levels despite a 50 % reduction in Asm activity, which could be explained with a (partial) inhibition of Ac. However, ceramide levels were not increased further by amitriptyline treatment, as you would expect based on the expedited death of Ac-deficient mice upon amitriptyline treatment.

Another possible explanation for the toxicity of amitriptyline in our Ac-deficient mouse model is the exacerbation of pre-existing organ injuries (i.e., hepatic disease) upon amitriptyline treatment. This is supported by the worsening of several clinical parameters in Ac-deficient mice upon amitriptyline treatment: Blood urea nitrogen levels, which are already elevated in Ac-deficient mice, were further increased upon amitriptyline treatment. Additionally, liver transaminases were also elevated further in treated mice, indicating hepatotoxic effects. However, other indicators of hepatic function were not changed by amitriptyline, such as bilirubin and albumin. Lactate dehydrogenase levels, indicating general cellular destruction, also worsened with treatment. Leukopenia was more pronounced, and platelet numbers were reduced by amitriptyline, indicating toxic effects to the bone marrow.

As mentioned earlier, it seems that amitriptyline-treated Ac-deficient mice drink less than controls. Elevated blood urea nitrogen levels and increased hematocrit are in line with this observation. However, it cannot be determined whether the treated mice drink less because of a taste avoidance due to amitriptyline in their drinking water, and the lack of fluid intake then makes them sicker or if consuming amitriptyline makes them sicker and they thus drink less due to their worsened state. In any case, exsiccosis cannot explain all toxic effects that were observed and further studies are necessary to identify the cause of death of Ac-deficient mice upon amitriptyline treatment.

In light of our successful genetic proof-of-concept study and the failure of the pharmacological study employing a functional inhibitor of Asm, our work suggests that effective and safe treatment of FD via Asm inhibition would require a direct inhibitor of Asm. Several such compounds have been described, including sphingomyelin analogs [45,46], natural products such as α-mangosteen [47,48], phosphoinositides [49,50], and bisphosphonates [51]. So far, all are ill-suited to widespread clinical application because they are either unspecific [45,48], unstable, exert biological effects themselves, or have poor distribution and membrane penetration [52]. Our genetic study provides proof-of-concept that a selective Asm inhibitor with widespread cellular distribution could ameliorate FD manifestations and meet the high clinical demand for a new treatment option for this disease.

## 4. Methods

### 4.1. Animal Husbandry

All mice were maintained on the C57BL/6-J background. Mice were bred and housed in the vivarium of the University Hospital Essen, Germany, under pathogen-free conditions as defined by the Federations of European Laboratory Animal Science Associations (FELASA). The pathogen-free status was routinely monitored through a sentinel program. Mice had *ad libitum* access to food and water and were kept on a 12 h/12 h light/dark cycle. Genotypes were assessed by PCR. Breeding and treatment of the mice were approved by the State Agency for Nature, Environment and Consumer Protection (LANUV NRW in Düsseldorf, Germany), approval number AZ 84-02.04.2015A.539.

### 4.2. Generation of the Asah1^tmEx1^ Smpd1^−/−^ Mouse Model

The generation of the *Asah1*^tmEx1^ mouse models and *Smpd1*^−/−^ model were previously described [20,53]. Since *Asah1*^tmEx1^ do not live long enough for breeding and the competent authority (LANUV NRW) only granted us permission to keep *Smpd1*^−/−^ until the age of 10 weeks, we initially bred heterozygous *Asah1*^tmEx1^ mice to heterozygous *Smpd1*^+/−^ mice to obtain *Asah1*^tmEx1^
*Smpd1*^+/−^ mice. We then bred these to each other to obtain all genotypes used in our study: Wt, *Asah1*^tmEx1^*, Asah1*^tmEx1^
*Smpd1*^+/−^ and *Asah1*^tmEx1^
*Smpd1*^−/−^ mice. Due to the poor ratios, however, we also used Wt and *Asah1*^tmEx1^ mice obtained from breeding heterozygous *Asah1*^tmEx1^ mice to each other, as well as Wt and *Smpd1*^−/−^ mice obtained from breeding *Smpd1*^+/−^ to each other and *Asah1*^tmEx1^
*Smpd1*^−/−^ mice obtained from breeding *Asah1*^tmEx1^
*Smpd1*^−/−^ mice with each other. No differences were observed between the respective genotypes based on their parentage.

### 4.3. Ceramide and Sphingomyelin Quantification

Ceramides and sphingomyelin were extracted from spleen tissue and quantified as recently described [54]. Briefly, lipid extraction was performed using C17-ceramide and C16-d31-sphingomyelin as internal standards (Avanti Polar Lipids, Alabaster, AL, USA). Sample analysis was carried out by liquid chromatography tandem-mass spectrometry (LC-MS/MS) a Quadrupole-Time of Flight 6530 mass spectrometer (Agilent Technologies, Waldbronn, Germany) operating in the positive ESI mode. The precursor ions of ceramides (C16-ceramide (*m*/*z* 520.508), C17-ceramide (*m*/*z* 534.524), C18-ceramide (*m*/*z* 548.540), C20-ceramide (*m*/*z* 576.571), C22-ceramide (*m*/*z* 604.602), C24-ceramide (*m*/*z* 632.634) and C24:1-ceramide (*m*/*z* 630.618)) were cleaved into the fragment ion *m*/*z* 264.270. The precursor ions of sphingomyelin (C16-sphingomyelin (*m*/*z* 703.575), C16-d_31_-sphingomyelin (*m*/*z* 734.762), C18-sphingomyelin (*m*/*z* 731.606), C20-sphingomyelin (*m*/*z* 759.638), C22-sphingomyelin (*m*/*z* 787.669), C24-sphingomyelin (*m*/*z* 815.700) and C24:1-sphingomyelin (*m*/*z* 813.684)) were cleaved into the fragment ion *m*/*z* 184.074. Quantification was performed with MassHunter Software (Agilent Technologies, Santa Clara, CA, USA). Determined sphingolipid amounts were normalized to the protein content of the tissue homogenate used for lipid extraction.

### 4.4. Histopathological Assessment

Mice were sacrificed by CO_2_ exposure and perfused with 0.9% sodium chloride, followed by 4% paraformaldehyde (PFA) via the right ventricle. Organs were dissected and fixed further in 4% PFA, dehydrated with ethanol, embedded in paraffin and trimmed to 4–6 µm thin sections. These were stained with hematoxylin and eosin and analyzed on a Leica TCS-SP5 confocal microscope (Leica Mikrosysteme Vertrieb GmbH, Wetzlar, Germany). Histiocytic infiltration, in various tissues and liver fibrosis were scored on a scale of 0 (no signs) to 4 (very severe) by a blinded investigator.

### 4.5. Cytokine Quantification

MCP-1, MIP-1alpha, IP-10, and VEGF levels were measured by enzyme-linked immunosorbent assay (ELISA). Mice were sacrificed by CO_2_ exposure, blood was drawn from the *V. cava inferior*, and serum was obtained by centrifugation at 2500× *g* for 20 min. Spleens were dissected, snap-frozen and homogenized in PBS + 10 µg/mL aprotinin (Sigma-Aldrich Chemie GmbH, Steinheim, Germany) + 10 µg/mL leupeptin (Sigma-Aldrich) + 1 × Complete Protease Inhibitor Cocktail (Roche, Freiburg, Germany) for 5 min at 50 Hz using a TissueLyser (Qiagen GmbH, Hilden, Germany). Samples were centrifuged for 5 min at 15,000× *g* to remove debris and cytokine levels were analyzed using ELISA kits according to the respective manufacturer’s instructions without any further treatment. ELISA kits for MCP1, MIP-1alpha, and VEGF levels were obtained from R&D (R&D Systems Minneapolis, MN, USA), and for IP-10 from Abcam (Abcam, Cambridge, UK).

### 4.6. Blood Counts and Clinical Chemistry Analysis

Blood was drawn and anticoagulated with EDTA for blood counts using a VetABC^TM^ (Scil, Viernheim, Germany). Serum was obtained by centrifugation at 2500× *g* from coagulated blood samples. Serum parameters were analyzed using a SpotChem^TM^ EZ chemistry analyzer with the corresponding parameter strips (Scil). When necessary, samples were diluted to enable the measurement of small sample volumes or when samples exceeded the maximum detection limit of a given parameter.

### 4.7. Amitriptyline Treatment

Amitriptyline hydrochloride (Sigma-Aldrich) was dissolved in normal saline (prepared with distilled water) at 180 mg/L. The fresh solution was prepared every three days. Mice received this as their drinking water, to which the mice had ad libitum access. This regimen has previously been reported to effectively inhibit Asm in vivo [35,36]. Control mice received only normal saline. We started treatment at four weeks of age (d28), shortly before the onset of weight loss in Ac-deficient mice. For determination of Asm activity and sphingolipid levels, mice were treated in this manner for seven days (analysis on d35).

### 4.8. Determination of Asm Activity

Asm activity was determined as recently described [55]. Briefly, mice were sacrificed, organs dissected, snap-frozen, and pulverized using a mortar and pestle filled with liquid nitrogen. A scoop of the organ powder was transferred to 2 mL reaction tubes containing a 5-mm stainless steel bead, which were pre-cooled on dry-ice. Samples were incubated at room temperature for 2 min prior to the addition of lysis buffer (250 nM sodium acetate, 1% NP-40, pH 5.0) and homogenization in a TissueLyser (Qiagen GmbH, Hilden, Germany) for 5 Min at 50 Hz. An aliquot was removed for protein determination, and another aliquot was dispensed on ice for determination of Asm activity. Asm lysis buffer was added to a final volume of 20 µL. Samples and the BODIPY-sphingomyelin substrate solution (BODIPY-sphingomyelin (Thermo Fisher Scientific, Waltham, MA, USA) diluted in 250 nM sodium acetate, 0.1% NP-40, pH 5.0) were sonicated in an ultrasonic bath for 10 min to induce micelle formation, before 100 pmol sonicated BODIPY-sphingomyelin was added to each sample. The reaction was carried out for 1 h at 37 °C, and then terminated by lipid extraction through addition of chloroform:methanol (2:1, *v*/*v*), vortexing and centrifugation of 5 min at 15,000× g. The lower phase was collected and dried in a SpeedVac at 37 °C. Dried lipids were resuspended in 20 µL chloroform:methanol (2:1, *v*/*v*) and spotted onto a thin layer chromatography (TLC) Silica G60 plate (Merck KGaA, Darmstadt, Germany). The TLC run was conducted with chloroform:methanol (80:20, *v*/*v*) as the running buffer. The plate was then dried and imaged using a Typhoon FLA 9500 (GE Healthcare Europe GmbH, Freiburg, Germany). Spots were quantified with ImageQuant software (GE Healthcare, ImageQuant TL 1D, Version 8.1).

### 4.9. Statistical Analyses

Data are presented as individual replicates with arithmetic means ± standard deviation. For multiple comparisons, we first confirmed the normal distribution of the data with D’Agostino and Pearson omnibus normality test. We then used analysis of variance (ANOVA) with the indicated posttests to test for significant differences between selected pairs. Histopathological scores were compared with the Wilcoxon signed-rank test and survival curves with the log-rank/Mantel-Cox test. All data were obtained from independent measurements.

## Figures and Tables

**Figure 1 ijms-20-06253-f001:**
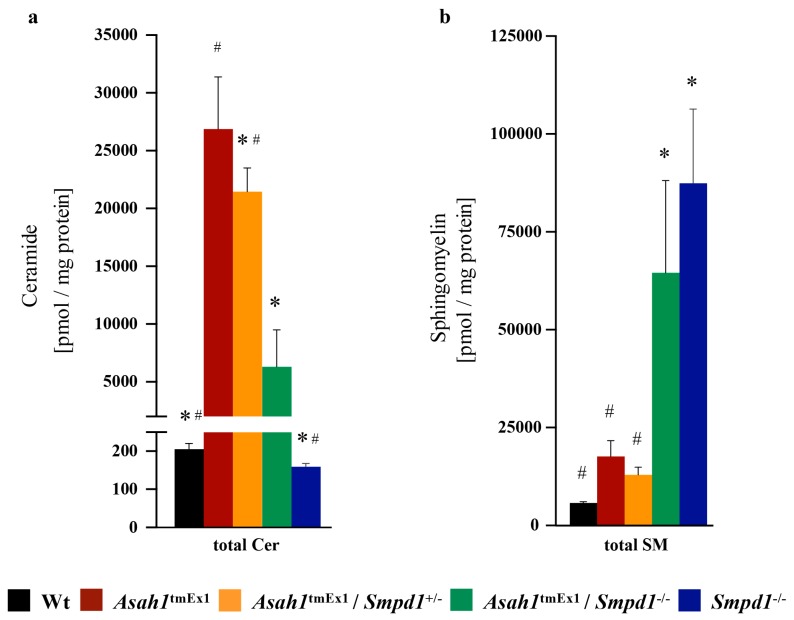
Ceramide accumulation in acid ceramidase deficient mice is blunted by acid sphingomyelinase co-ablation (**a**,**b**) The effects of Asm co-deficiency on ceramide (**a**) and sphingomyelin (**b**) levels in Ac deficient mice were analyzed by liquid chromatography tandem-mass spectrometry (LC-MS/MS) of snap-frozen spleen samples. * *p* < 0.05 compared to *Asah1*^tmEx1^, ^#^
*p* < 0.05 compared to *Asah1*^tmEx1^/*Smpd1*^−/−^ (ANOVA with Dunnett posttests). The legend at the bottom of the figure applies to both (**a**) and (**b**).

**Figure 2 ijms-20-06253-f002:**
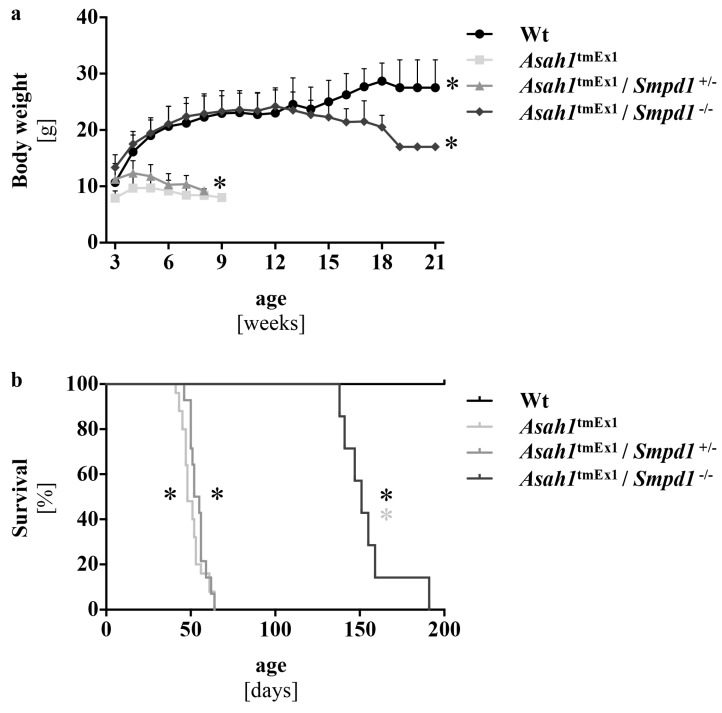
Asm co-ablation improves weight gain and prolongs survival of Ac deficient mice (**a**). Body weight of different genotypes. Starting after weaning, mice were weighted once a week. Data are presented as mean ± SD (*n* > 14 mice for each group). * *p* < 0.05 compared to *Asah1*^tmEx1^ (repeated measures ANOVA with Bonferroni posttests). (**b**) Survival curves of different genotypes (*n* ≥ 7 for each group). Mice were inspected daily to monitor survival. * *p* < 0.05 compared to Wt (black) or *Asah1*^tmEx1^ (grey) (log-rank/Mantel-Cox test).

**Figure 3 ijms-20-06253-f003:**
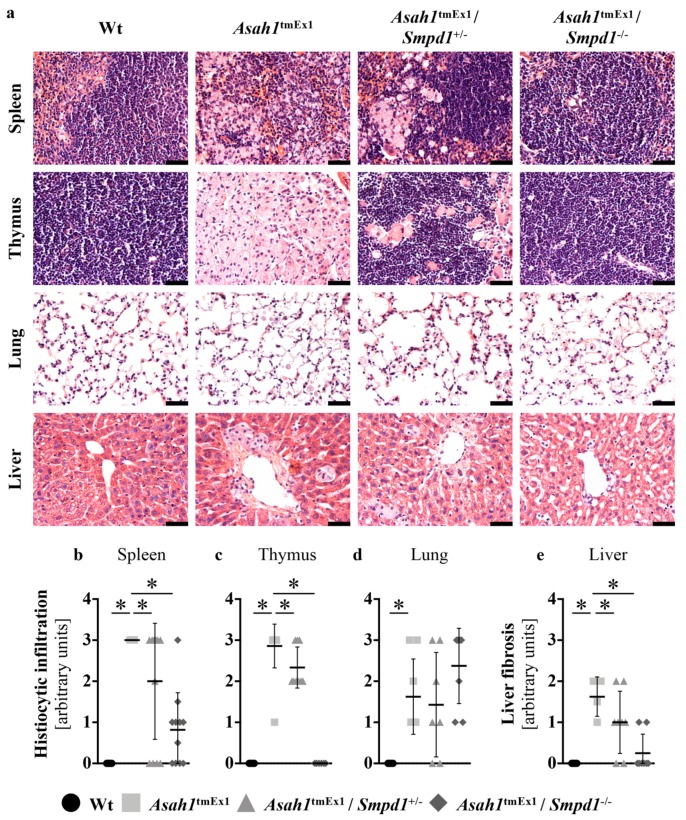
Asm co-ablation ameliorates histopathological signs of Farber disease (**a**) Histopathology of the effects of ceramide and sphingomyelin accumulation. Hematoxylin/eosin staining of perfused, paraformaldehyde-fixed and paraffin-embedded tissue sections. Representative images of *n* > 6 mice are shown for each group. Scale bar: 50 μm. (**b**–**e**) Histiocytic infiltration with foamy macrophages (**b**–**d**) and liver fibrosis (**e**) were scored on a scale of 0 (no signs) to 4 (very severe) in spleen (**b**), thymus (**c**), lung (**d**) and liver (**e**) by a blinded investigator. * *p* < 0.05 compared to *Asah1*^tmEx1^ mice (Wilcoxon signedrank test). The legend at the bottom of the figures applies to figures (**b**) through (**e**).

**Figure 4 ijms-20-06253-f004:**
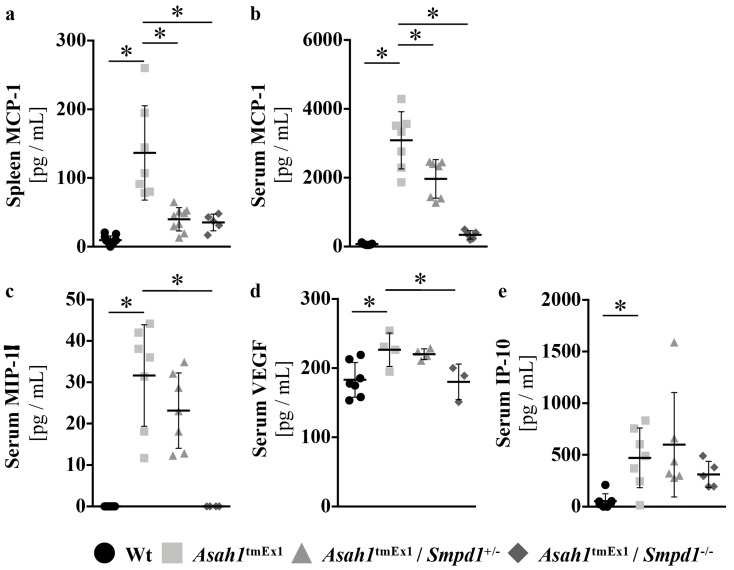
Asm ablation in Ac deficient mice improves the cytokine profile characteristic for FD (**a**–**e**) Spleen (**a**) and serum (**b**–**e**) cytokine levels were quantified by enzyme-linked immunosorbent assay (ELISA). * *p* < 0.05 compared to *Asah1*^tmEx1^ mice (ANOVA with Dunnett posttests). The legend at the bottom applies to all figures.

**Figure 5 ijms-20-06253-f005:**
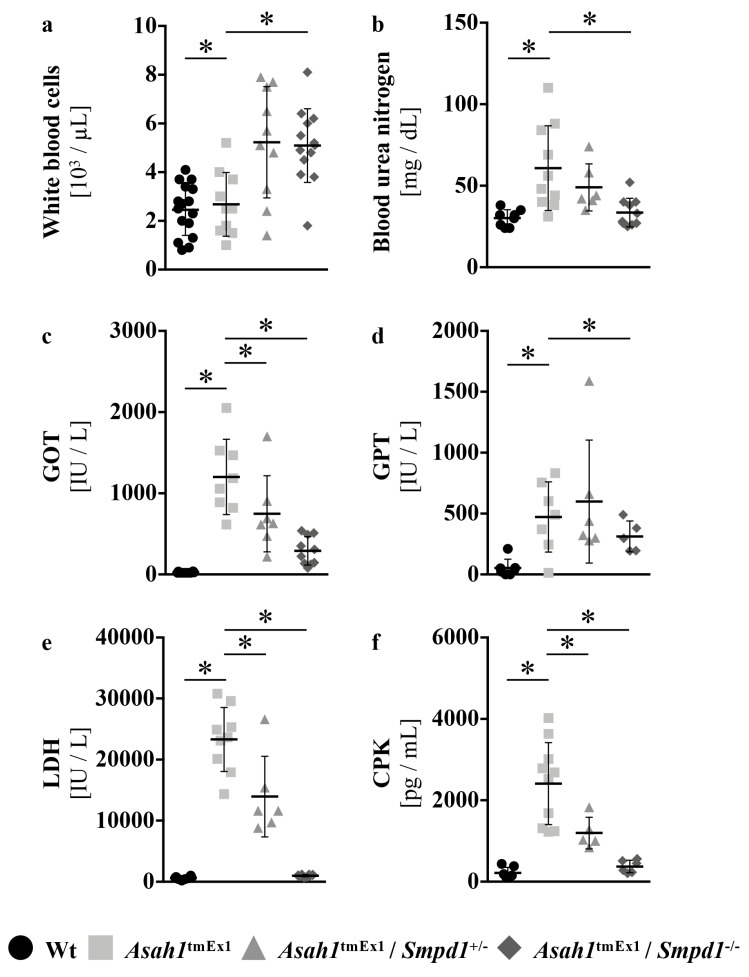
Asm co-ablation in Ac deficient mice improves clinical parameters to assess overall health (**a**) Leukocyte numbers in peripheral blood anti-coagulated with EDTA were determined using a VetABCTM (Scil). * *p* < 0.05 compared to *Asah1*^tmEx1^ mice (ANOVA with Dunnett posttests). (**b**–**f**) Clinical parameters were determined in serum using a SpotchemTM (Scil). GOT: glutamic-oxaloacetic transaminase, GPT: glutamate-pyruvate transaminase, LDH: lactate dehydrogenase, CPK: creatinine phosphokinase. * *p* < 0.05 compared to *Asah1*^tmEx1^ mice (ANOVA with Dunnett posttests). The legend at the bottom applies to all figures.

**Figure 6 ijms-20-06253-f006:**
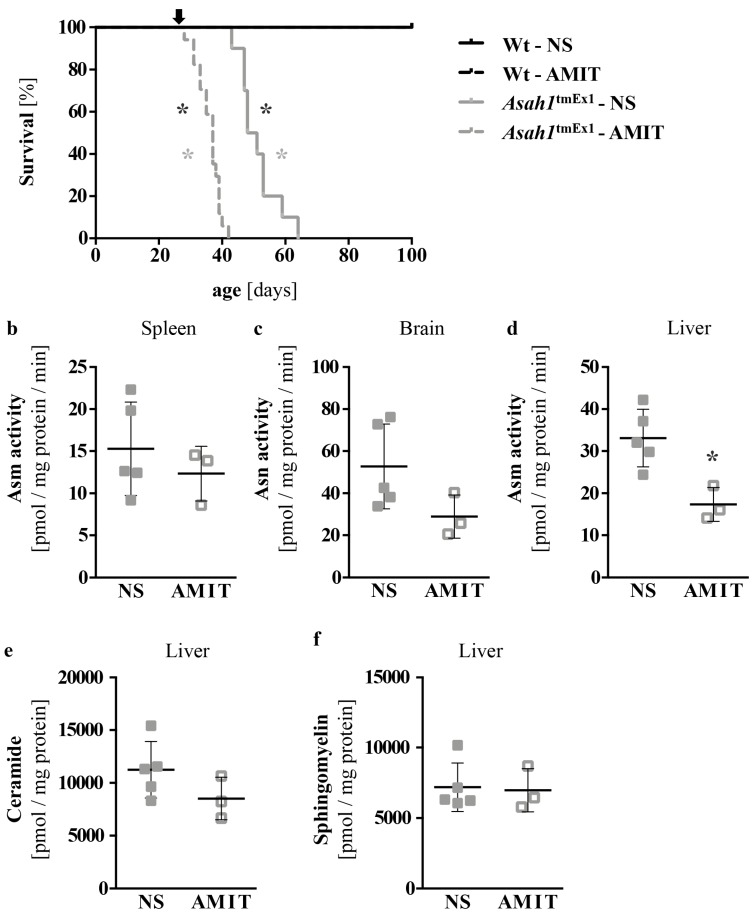
Pharmacological targeting of Asm with Amitriptyline kills Ac deficient mice. Mice were treated with 180 mg/L amitriptyline (AMIT) in normal saline via their drinking water starting at 4 weeks of age (d28, indicated by arrow). Controls received normal saline (NS). (**a**) Survival was monitored daily. *n* = 10–17 mice/group. * *p* < 0.05 compared to Wt-NS (black) or *Asah1*^tmEx1^-NS (grey) (logrank/Mantel-Cox test). (**b**–**d**) In a second cohort of mice, Asm activity was determined on d35 in spleen, brain and liver of *Asah1*^tmEx1^ mice that survived the first seven days of treatment. *n* = 3–5 mice/group. * *p* < 0.05 compared to NS (Student t test). (**e**–**f**) In that same cohort, total ceramide and sphingomyelin levels in the liver were determined by liquid chromatography tandem-mass spectrometry (LC-MS/MS). No significant differences were detected compared to *Asah1*^tmEx1^-NS (Student *t* test).

**Figure 7 ijms-20-06253-f007:**
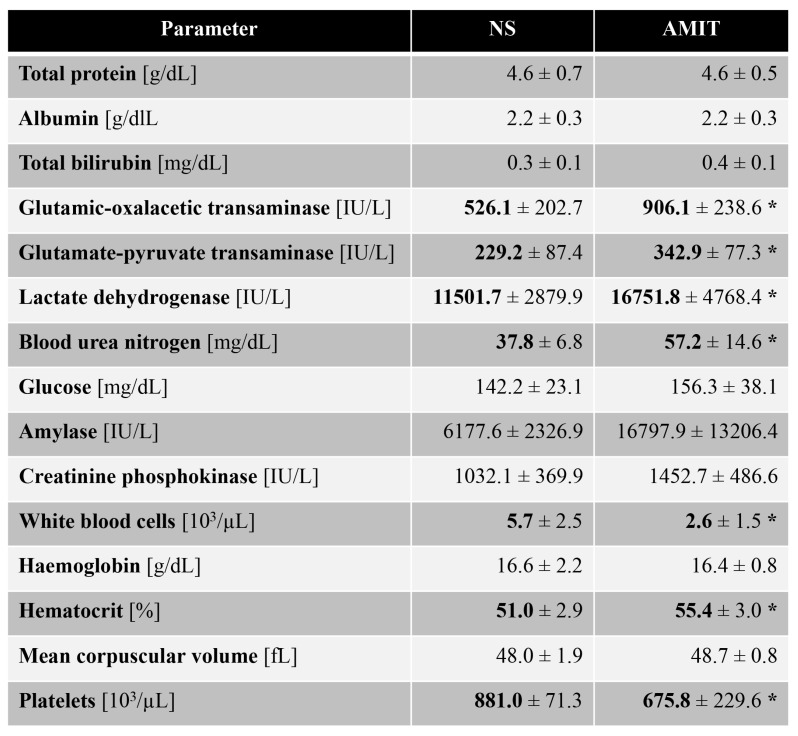
Amitryptiline treatment of Ac deficient mice shows toxic effects in blood analyses. Blood counts in peripheral blood anti-coagulated with EDTA were determined using a VetABCTM (Scil). * *p* < 0.05 compared to *Asah1*^tmEx1^ mice (ANOVA with Dunnett posttests).

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
