# Peer review of "Acid Sphingomyelinase Deficiency Ameliorates Farber Disease"

_ijms, 2019, doi:10.3390/ijms20246253_

Round 1
Reviewer 1 Report
This is an excellent article on and animal model development and therapeutics by Beckmann et al.
Overall the manuscript delves in depth into the ceramide pathway and the basis on which the animal model has been developed for a rare disease like Farber disease which has no definitive treatment available.
The study provides a first proof-of-concept that acid sphingomyelinase could be a potential new therapeutic target for Farber disease to reduce disease manifestations and prolong survival.
The following modifications/ experiments would further improve the manuscript before acceptance.
Major comments:
Page 4, line 169: Pharmacological targeting of Asm with amitriptyline kills Ac deficient mice. This is a very significant finding which though negative at this stage could provide vital data.It would be nice to develop the data further by the following experiments/clarifications.
Levels of ceramide, and Sphingomyelin in these animals measured in the lung tissue and plasma Details as to how many experiments were done? Survival curve with details such as date of beginning of experiment and mortality. What was the dose of amitriptyline given? Any dose range tried and titrated as per sphingomyelin levels. Dose response curve and correlation with ceramide and sphingomyelin levels.The results could be extremely helpful. This could help the future experimental design and set the stage for large animal studies. It will be a shame to set aside the valuable study as meaningless after having done so much of excellent work.
Minor comments
Page 1, line 22 which reads “Acid sphingomyelinase is the enzyme that generates ceramide upstream of acid 22 ceramidase in the lysosomes”. Please move this sentence prior to the previous one as it introduces the enzyme better and the sequence flows better. Page 3, Line 109: “Only C20 and C22 108 ceramide levels were not ameliorated by homozygous Asm co-deficiency”. An explanation for the differential expression of various ceramide species or a similar varied expression in disease condition would explain the findings better or correlate the same with other pathological conditions.Author Response
We would like to thank both reviewers for their time and input!
Page 4, line 169: Pharmacological targeting of Asm with amitriptyline kills Ac deficient mice. This is a very significant finding which though negative at this stage could provide vital data.
It would be nice to develop the data further by the following experiments/clarifications.
Levels of ceramide, and Sphingomyelin in these animals measured in the lung tissue and plasma Details as to how many experiments were done? Survival curve with details such as date of beginning of experiment and mortality. What was the dose of amitriptyline given? Any dose range tried and titrated as per sphingomyelin levels. Dose response curve and correlation with ceramide and sphingomyelin levels.”
The experimental details (dose of amitriptyline, beginning of treatment, number of treated animals, etc) can be found in the Figure Legends and Materials and Methods section. We have updated the manuscript to make this clearer. The administered dose of amitriptyline was established previously and has been repeatedly reported in the literature. Based on our experience, it constitutes a fairly high dose of amitriptyline, to account for the high sphingolipid levels found in Ac-deficient mice. However, we did not titrate the dose based on the sphingomyelin levels in the Ac-deficient mice. While a dose response curve in correlation to the sphingolipid level would certainly be very interesting, we do not have approval from the competent authority (LANUV NRW) do conduct these experiments. In light of the toxicity of amitriptyline treatment in Ac deficient mice that we’ve observed, it is also highly unlikely that such a study would be approved.
We have addressed your comments as much as possible within the scope of our approved experiments. We have added new data regarding ceramide and sphingomyelin levels and clinical parameters in Ac deficient animals that were subjected to the approved amitriptyline treatment (please refer to updated Figure 6 and Supplementary Figure 2). Among the tested organs, a significant decrease of Asm activity was only observed for the liver, with an approximately 50 % reduction. This did not translate into significant changes in ceramide or sphingomyelin levels in the liver. Based on our experiments and previous reports using the same dose of amitriptyline, it was surprising not to find a significant reduction of Asm activity in all tested organs. This indicates that the Ac deficient mice may not have consumed as much amitriptyline with their drinking water as healthy controls, which is supported by an increased hematocrit and worsened kidney function in these mice. At this point, it is unclear if the Ac deficient mice avoided amitriptyline due to the taste – something that has not been reported as an issue with healthy mice – or if they generally drink less due to their diseased state. In any case, we also observed further liver- and bone marrow-toxic effects of amitriptyline in Ac deficient mice, which cannot be explained by exsiccosis alone. However, future studies are still necessary to determine cause of death upon amitriptyline treatment.
Page 1, line 22 which reads “Acid sphingomyelinase is the enzyme that generates ceramide upstream of acid 22 ceramidase in the lysosomes”. Please move this sentence prior to the previous one as it introduces the enzyme better and the sequence flows better.
Thank you for your suggestions. We have altered the text accordingly.
Page 3, Line 109: “Only C20 and C22 108 ceramide levels were not ameliorated by homozygous Asm co-deficiency”. An explanation for the differential expression of various ceramide species or a similar varied expression in disease condition would explain the findings better or correlate the same with other pathological conditions.
Unfortunately, nothing is conclusively known about the contribution of each individual ceramide species to the disease phenotype at this point. There are also no reports that detail the expression of the different ceramide species in correlation to disease severity.

Reviewer 2 Report
After being provided the figures, I see the results are significant and match the results and interpretation provided. I think the work has made a mechanistic advance in the field and demonstrates a clear mechanistic link between an acid sphingomyelinase inhibitor and potential toxicity in Farber disease (and perhaps other diseases too). Figure 1 is hard to interpret with respect to significance and what the takeaway message is with respect to changes in ceramide. I recommend the figure be edited to clearly highlight the most important changes in one panel and perhaps the changes that are of little consequence in another. I think this will help the reader more quickly grasp the main changes.
Author Response
We would like to thank both reviewers for their time and input!
Figure 1 is hard to interpret with respect to significance and what the takeaway message is with respect to changes in ceramide. I recommend the figure be edited to clearly highlight the most important changes in one panel and perhaps the changes that are of little consequence in another. I think this will help the reader more quickly grasp the main changes.
As detailed in the previous response, the contribution of each individual ceramide species to the pathological manifestations of Farber Disease is currently unknown. Thus, we cannot say with certainty which changes are the most relevant. In order to acknowledge your concern regarding the presentation of Figure 1, however, we have limited Figure 1 to only depict total ceramide and sphingomyelin levels. We have moved the data on the individual sphingolipid species to supplemental figures.

Round 2
Reviewer 1 Report
The authors have addressed all the concerns satisfactorily